# A General Framework for Symmetric Property Estimation

**Moses Charikar**
Stanford University
moses@cs.stanford.edu

**Kirankumar Shiragur**
Stanford University
shiragur@stanford.edu

**Aaron Sidford**
Stanford University
sidford@stanford.edu

## Abstract

In this paper we provide a general framework for estimating symmetric properties of distributions from i.i.d. samples. For a broad class of symmetric properties we identify the *easy* region where empirical estimation works and the *difficult* region where more complex estimators are required. We show that by approximately computing the profile maximum likelihood (PML) distribution [ADOS16] in this difficult region we obtain a symmetric property estimation framework that is sample complexity optimal for many properties in a broader parameter regime than previous universal estimation approaches based on PML. The resulting algorithms based on these *pseudo PML distributions* are also more practical.

## 1 Introduction

Symmetric property estimation is a fundamental and well studied problem in machine learning and statistics. In this problem, we are given $n$ i.i.d samples from an unknown distribution[1] $\mathbf{p}$ and asked to estimate $\mathbf{f}(\mathbf{p})$, where $\mathbf{f}$ is a symmetric property (i.e. it does not depend on the labels of the symbols). Over the past few years, the computational and sample complexities for estimating many symmetric properties have been extensively studied. Estimators with optimal sample complexities have been obtained for several properties including entropy [VV11b, WY16a, JVHW15], distance to uniformity [VV11a, JHW16], and support [VV11b, WY15].

All aforementioned estimators were property specific and therefore, a natural question is to design a universal estimator. In [ADOS16], the authors showed that the distribution that maximizes the profile likelihood, i.e. the likelihood of the multiset of frequencies of elements in the sample, referred to as *profile maximum likelihood (PML) distribution*, can be used as a universal plug-in estimator. [ADOS16] showed that computing the symmetric property on the PML distribution is sample complexity optimal in estimating support, support coverage, entropy and distance to uniformity within accuracy $\epsilon > \frac{1}{n^{0.2499}}$. Further, this also holds for distributions that approximately optimize the PML objective with the approximation factor affecting the values of $\epsilon$ for which it holds.

Acharya et al. [ADOS16] posed two important and natural open questions. The first was to give an efficient algorithm for finding an approximate PML distribution, which was recently resolved in [CSS19]. The second open question is whether PML is sample competitive in all regimes of the accuracy parameter $\epsilon$? In this work, we make progress towards resolving this open question.

Firstly, we show that the PML distribution based plug-in estimator achieves optimal sample complexity for all $\epsilon$ for the problem of estimating support size. Next, we introduce a variation of the PML distribution that we call the *pseudo PML distribution*. Using this, we give a general framework for estimating a symmetric property. For entropy and distance to uniformity, this pseudo PML based framework achieves optimal sample complexity for a broader regime of the accuracy parameter than was known for the vanilla PML distribution.

We provide a general framework that could, in principle be applied to estimate any separable symmetric property $\mathbf{f}$, meaning $\mathbf{f}(\mathbf{p})$ can be written in the form of $\sum_{x \in \mathcal{D}} \mathbf{f}(\mathbf{p}_x)$. This motivation behind this framework is: for any symmetric property $\mathbf{f}$ that is separable, the estimate for $\mathbf{f}(\mathbf{p})$ can be split into two parts: $\mathbf{f}(\mathbf{p}) = \sum_{x \in B} \mathbf{f}(\mathbf{p}_x) + \sum_{x \in G} \mathbf{f}(\mathbf{p}_x)$, where $B$ and $G$ are a (property dependent) disjoint partition of the domain $\mathcal{D}$. We refer to $G$ as the good set and $B$ as the bad set. Intuitively, $G$ is the subset of domain elements whose contribution to $\mathbf{f}(\mathbf{p})$ is easy to estimate, i.e a simple estimator such as empirical estimate (with correction bias) works. For many symmetric properties, finding an appropriate partition of the domain is often easy. Many estimators in the literature [JVHW15, JHW16, WY16a] make such a distinction between domain elements. The more interesting and difficult case is estimating the contribution of the bad set: $\sum_{x \in B} \mathbf{f}(\mathbf{p}_x)$. Much of the work in these estimators is dedicated towards estimating this contribution using sophisticated techniques such as polynomial approximation. Our work gives a unified approach to estimating the contribution of the bad set. We propose a PML based estimator for estimating $\sum_{x \in B} \mathbf{f}(\mathbf{p}_x)$. We show that computing the PML distribution only on the set $B$ is sample competitive for entropy and distance to uniformity for almost all interesting parameter regimes thus (partially) handling the open problem proposed in [ADOS16]. Additionally, requiring that the PML distribution be computed on a subset $B \subseteq \mathcal{D}$ reduces the input size for the PML subroutine and results in practical algorithms (See Section 6).

To summarize, the main contributions of our work are:

- We make progress on an open problem of [ADOS16] on broadening the range of error parameter $\epsilon$ that one can obtain for universal symmetric property estimation via PML.

- We give a general framework for applying PML to new symmetric properties.

- As a byproduct of our framework, we obtain more practical algorithms that invoke PML on smaller inputs (See Section 6).

## 1.1 Related Work

For many natural properties, there has been extensive work on designing efficient estimators both with respect to computational time and sample complexity [HJWW17, HJM17, AOST14, RVZ17, ZVV+16, WY16b, RRSS07, WY15, OSW16, VV11b, WY16a, JVHW15, JHW16, VV11a]. We define and state the optimal sample complexity for estimating support, entropy and distance to uniformity. For entropy, we also discuss the regime in which the empirical distribution is sample optimal.

**Entropy:** For any distribution $\mathbf{p} \in \Delta^{\mathcal{D}}$, the entropy $H(\mathbf{p}) \stackrel{\text{def}}{=} -\sum_{x \in \mathcal{D}} \mathbf{p}_x \log \mathbf{p}_x$. For $\epsilon \geq \frac{\log N}{N}$ (the interesting regime), where $N \stackrel{\text{def}}{=} |\mathcal{D}|$, the optimal sample complexity for estimating $H(\mathbf{p})$ within additive accuracy $\epsilon$ is $O(\frac{N}{\log N} \frac{1}{\epsilon})$ [WY16a]. Further if $\epsilon < \frac{\log N}{N}$, then [WY16a] showed that empirical distribution is optimal.

**Distance to uniformity:** For any distribution $\mathbf{p} \in \Delta^{\mathcal{D}}$, the distance to uniformity $\|\mathbf{p} - u\|_1 \stackrel{\text{def}}{=} \sum_{x \in \mathcal{D}} |\mathbf{p}_x - \frac{1}{N}|$, where $u$ is the uniform distribution over $\mathcal{D}$. The optimal sample complexity for estimating $\|\mathbf{p} - u\|_1$ within additive accuracy $\epsilon$ is $O(\frac{N}{\log N} \frac{1}{\epsilon^2})$ [VV11a, JHW16].

**Support:** For any distribution $\mathbf{p} \in \Delta^{\mathcal{D}}$, the support of distribution $S(\mathbf{p}) \stackrel{\text{def}}{=} |\{x \in \mathcal{D} \mid \mathbf{p}_x > 0\}|$. Estimating support is difficult in general because we need sufficiently large number of samples to observe elements with small probability values. Suppose for all $x \in \mathcal{D}$, if $\mathbf{p}_x \in \{0\} \cup [\frac{1}{k}, 1]$, then [WY15] showed that the optimal sample complexity for estimating support within additive accuracy $\epsilon k$ is $O(\frac{k}{\log k} \log^2 \frac{1}{\epsilon})$.

PML was introduced by Orlitsky et al. [OSS+04] in 2004. The connection between PML and universal estimators was first studied in [ADOS16]. As discussed in the introduction, PML based plug-in estimator applies to a restricted regime of error parameter $\epsilon$. There have been several other approaches for designing universal estimators for symmetric properties. Valiant and Valiant [VV11b] adopted and rigorously analyzed a linear programming based approach for universal estimators proposed by [ET76] and showed that it is sample complexity optimal in the constant error regime for estimating certain symmetric properties (namely, entropy and support size). Recent work of Han et al. [HJW18] applied a local moment matching based approach in designing efficient universal

symmetric property estimators for a single distribution. [HJW18] achieves the optimal sample complexity in restricted error regimes for estimating the power sum function, support and entropy.

Recently, [YOSW18] gave a different unified approach to property estimation. They devised an estimator that uses $n$ samples and achieves the performance attained by the empirical estimator with $n\sqrt{\log n}$ samples for a wide class of properties and for all underlying distributions. This result is further strengthened to $n \log n$ samples for Shannon entropy and a broad class of other properties including $\ell_1$-distance in [HO19b].

Independently of our work, authors in [HO19a] propose *truncated PML* that is slightly different but similar in the spirit to our idea of pseudo PML; refer [HO19a] for further details.

## 1.2 Organization of the Paper

In Section 2 we provide basic notation and definitions. We present our general framework in Section 3 and state all our main results. In Section 4, we provide proofs of the main results of our general framework. In Section 5, we use these results to establish the sample complexity of our estimator in the case of entropy (See Section 5.1) and distance to uniformity (See Section 5.2). Due to space constraints, many proofs are deferred to the appendix. In Section 6, we provide experimental results for estimating entropy using pseudo PML and other state-of-the-art estimators. Here we also demonstrate the practicality of our approach.

## 2 Preliminaries

Let $[a]$ denote all integers in the interval $[1, a]$. Let $\Delta^{\mathcal{D}} \subset [0, 1]_{\mathbb{R}}^{\mathcal{D}}$ be the set of all distributions supported on domain $\mathcal{D}$ and let $N$ be the size of the domain. Throughout this paper we restrict our attention to discrete distributions and assume that we receive a sequence of $n$ independent samples from an underlying distribution $\mathbf{p} \in \Delta^{\mathcal{D}}$. Let $\mathcal{D}^n$ be the set of all length $n$ sequences and $y^n \in \mathcal{D}^n$ be one such sequence with $y_i^n$ denoting its $i$th element. The probability of observing sequence $y^n$ is:

$$\mathbb{P}(\mathbf{p}, y^n) \stackrel{\text{def}}{=} \prod_{x \in \mathcal{D}} \mathbf{p}_x^{\mathbf{f}(y^n, x)}$$

where $\mathbf{f}(y^n, x) = |\{i \in [n] \mid y_i^n = x\}|$ is the frequency/multiplicity of symbol $x$ in sequence $y^n$ and $\mathbf{p}_x$ is the probability of domain element $x \in \mathcal{D}$. We next formally define profile, PML distribution and approximate PML distribution.

**Definition 2.1** (Profile). For a sequence $y^n \in \mathcal{D}^n$, its *profile* denoted $\phi = \Phi(y^n) \in \mathbb{Z}_+^n$ is $\phi \stackrel{\text{def}}{=} (\phi(j))_{j \in [n]}$ where $\phi(j) \stackrel{\text{def}}{=} |\{x \in \mathcal{D} | \mathbf{f}(y^n, x) = j\}|$ is the number of domain elements with frequency $j$ in $y^n$. We call $n$ the length of profile $\phi$ and use $\Phi^n$ denote the set of all profiles of length $n$. [2]

For any distribution $\mathbf{p} \in \Delta^{\mathcal{D}}$, the probability of a profile $\phi \in \Phi^n$ is defined as:

$$\mathbb{P}(\mathbf{p}, \phi) \stackrel{\text{def}}{=} \sum_{\{y^n \in \mathcal{D}^n \mid \Phi(y^n) = \phi\}} \mathbb{P}(\mathbf{p}, y^n) \tag{1}$$

The distribution that maximizes the probability of a profile $\phi$ is the profile maximum likelihood distribution and we formally define it next.

**Definition 2.2** (Profile maximum likelihood distribution). For any profile $\phi \in \Phi^n$, a *Profile Maximum Likelihood* (PML) distribution $\mathbf{p}_{pml,\phi} \in \Delta^{\mathcal{D}}$ is: $\mathbf{p}_{pml,\phi} \in \arg\max_{\mathbf{p} \in \Delta^{\mathcal{D}}} \mathbb{P}(\mathbf{p}, \phi)$ and $\mathbb{P}(\mathbf{p}_{pml,\phi}, \phi)$ is the maximum PML objective value. Further, a distribution $\mathbf{p}_{pml,\phi}^{\beta} \in \Delta^{\mathcal{D}}$ is a $\beta$-*approximate PML* distribution if $\mathbb{P}(\mathbf{p}_{pml,\phi}^{\beta}, \phi) \geq \beta \cdot \mathbb{P}(\mathbf{p}_{pml,\phi}, \phi)$.

We next provide formal definitions for separable symmetric property and an estimator.

**Definition 2.3** (Separable Symmetric Property). A symmetric property $\mathbf{f} : \Delta^{\mathcal{D}} \to \mathbb{R}$ is separable if for any $\mathbf{p} \in \Delta^{\mathcal{D}}$, $f(\mathbf{p}) \stackrel{\text{def}}{=} \sum_{x \in \mathcal{D}} \mathbf{g}(\mathbf{p}_x)$, for some function $\mathbf{g} : \mathbb{R} \to \mathbb{R}$. Further for any subset $S \subset \mathcal{D}$, we define $f_S(\mathbf{p}) \stackrel{\text{def}}{=} \sum_{x \in S} \mathbf{g}(\mathbf{p}_x)$.

**Definition 2.4.** A property estimator is a function $\hat{\mathbf{f}} : \mathcal{D}^n \to \mathbb{R}$, that takes as input $n$ samples and returns the estimated property value. The sample complexity of $\hat{\mathbf{f}}$ for estimating a symmetric property $\mathbf{f}(\mathbf{p})$ is the number of samples needed to estimate $\mathbf{f}$ up to accuracy $\epsilon$ and with constant probability. The optimal sample complexity of a property $\mathbf{f}$ is the minimum number of samples of any estimator.

## 3 Main Results

As discussed in the introduction, one of our motivations was to provide a better analysis for the PML distribution based plug-in estimator. In this direction, we first show that the PML distribution is sample complexity optimal in estimating support in all parameter regimes. Estimating support is difficult in general and all previous works make the assumption that the minimum non-zero probability value of the distribution is at least $\frac{1}{k}$. In our next result, we show that the PML distribution under this constraint is sample complexity optimal for estimating support.

**Theorem 3.1.** *The PML distribution* [3] *based plug-in estimator is sample complexity optimal in estimating support for all regimes of error parameter $\epsilon$.*

For support, we show that an approximate PML distribution is sample complexity optimal as well.

**Theorem 3.2.** *For any constant $\alpha > 0$, an $\exp(-\epsilon^2 n^{1-\alpha})$-approximate PML distribution* [3] *based plug-in estimator is sample complexity optimal in estimating support for all regimes of error $\epsilon$.*

We defer the proof of both these theorems to Appendix A.

For entropy and distance to uniformity, we study a variation of the PML distribution we call the pseudo PML distribution and present a general framework for symmetric property estimation based on this. We show that this pseudo PML based general approach gives an estimator that is sample complexity optimal for estimating entropy and distance to uniformity in broader parameter regimes. To motivate and understand this general framework we first define new generalizations of the profile, PML and approximate PML distributions.

**Definition 3.3** (*S*-pseudo Profile). For any sequence $y^n \in \mathcal{D}^n$ and $S \subseteq \mathcal{D}$, its *S-pseudo* profile denoted $\phi_S = \Phi_S(y^n)$ is $\phi \stackrel{\text{def}}{=} (\phi_S(j))_{j \in [n]}$ where $\phi_S(j) \stackrel{\text{def}}{=} |\{x \in S \mid \mathbf{f}(y^n, x) = j\}|$ is the number of domain elements in $S$ with frequency $j$ in $y^n$. We call $n$ the length of $\phi_S$ as it represents the length of the sequence $y^n$ from which this pseudo profile was constructed. Let $\Phi_S^n$ denote the set of all *S*-pseudo profiles of length $n$.

For any distribution $\mathbf{p} \in \Delta^{\mathcal{D}}$, the probability of a *S*-pseudo profile $\phi_S \in \Phi_S^n$ is defined as:

$$\mathbb{P}(\mathbf{p}, \phi_S) \stackrel{\text{def}}{=} \sum_{\{y^n \in \mathcal{D}^n \;|\; \Phi_S(y^n) = \phi_S\}} \mathbb{P}(\mathbf{p}, y^n) \tag{2}$$

We next define the *S*-pseudo PML and $(\beta, S)$-approximate pseudo PML distributions that are analogous to the PML and approximate PML distributions.

**Definition 3.4** (*S*-pseudo PML distribution). For any *S*-pseudo profile $\phi_S \in \Phi_S^n$, a distribution $\mathbf{p}_{\phi_S} \in \Delta^{\mathcal{D}}$ is a *S-pseudo PML* distribution if $\mathbf{p}_{\phi_S} \in \arg\max_{\mathbf{p} \in \Delta^{\mathcal{D}}} \mathbb{P}(\mathbf{p}, \phi_S)$.

**Definition 3.5** (($\beta, S$)-approximate pseudo PML distribution). For any profile $\phi_S \in \Phi_S^n$, a distribution $\mathbf{p}_{\phi_S}^{\beta} \in \Delta^{\mathcal{D}}$ is a $(\beta, S)$-*approximate pseudo PML* distribution if $\mathbb{P}(\mathbf{p}_{\phi_S}^{\beta}, \phi_S) \geq \beta \cdot \mathbb{P}(\mathbf{p}_{\phi_S}^{\beta}, \phi_S)$.

For notational convenience, we also define the following function.

**Definition 3.6.** For any subset $S \subseteq \mathcal{D}$, the function $\text{Freq} : \Phi_S^n \to 2^{\mathbb{Z}_+}$ takes input a *S*-pseudo profile and returns the set with all distinct frequencies in $\phi_S$.

Using the definitions above, we next give an interesting generalization of Theorem 3 in [ADOS16].

**Theorem 3.7.** *For a symmetric property $\mathbf{f}$ and $S \subseteq \mathcal{D}$, suppose there is an estimator $\hat{\mathbf{f}} : \Phi_S^n \to \mathbb{R}$, such that for any $\mathbf{p}$ and $\phi_S \sim \mathbf{p}$ the following holds,*

$$\mathbb{P}\left(|\mathbf{f}_S(\mathbf{p}) - \hat{\mathbf{f}}(\phi_S)| \geq \epsilon\right) \leq \delta ,$$

*then for any* $F \in 2^{\mathbb{Z}+}$, *a* $(\beta, S)$-*approximate pseudo PML distribution* $\boldsymbol{p}^{\beta}_{\phi_S}$ *satisfies:*

$$\mathbb{P}\left(|\boldsymbol{f}_S(\boldsymbol{p}) - \boldsymbol{f}_S(\boldsymbol{p}^{\beta}_{\phi_S})| \geq 2\epsilon\right) \leq \frac{\delta n^{|F|}}{\beta} \mathbb{P}\left(\text{Freq}\left(\phi_S\right) \subseteq F\right) + \mathbb{P}\left(\text{Freq}\left(\phi_S\right) \not\subseteq F\right) .$$

Note that in the theorem above, the error probability with respect to a pseudo PML distribution based estimator has dependency on $\frac{\delta n^{|F|}}{\beta}$ and $\mathbb{P}\left(\text{Freq}\left(\phi_S\right) \not\subseteq F\right)$. However Theorem 3 in [ADOS16] has error probability $\frac{\delta e^{\sqrt{n}}}{\beta}$. This is the bottleneck in showing that PML works for all parameter regimes and the place where pseudo PML wins over the vanilla PML based estimator, getting non-trivial results for entropy and distance to uniformity. We next state our general framework for estimating symmetric properties. We use the idea of sample splitting which is now standard in the literature [WY16a, JVHW15, JHW16, CL11, Nem03].

---

**Algorithm 1** General Framework for Symmetric Property Estimation

---

1: **procedure** PROPERTY ESTIMATION($x^{2n}, \mathbf{f}, F$)
2:　　Let $x^{2n} = (x_1^n, x_2^n)$, where $x_1^n$ and $x_2^n$ represent first and last $n$ samples of $x^{2n}$ respectively.
3:　　Define $S \overset{\text{def}}{=} \{y \in \mathcal{D} \mid f(x_1^n, y) \in F\}$.
4:　　Construct profile $\phi_S$, where $\phi_S(j) \overset{\text{def}}{=} |\{y \in S \mid \mathbf{f}(x_2^n, y) = j\}|$.
5:　　Find a $(\beta, S)$-approximate pseudo PML distribution $\mathbf{p}^{\beta}_{\phi_S}$ and empirical distribution $\hat{\mathbf{p}}$ on $x_2^n$.
6:　　**return** $\mathbf{f}_S(\mathbf{p}^{\beta}_{\phi_S}) + \mathbf{f}_{\bar{S}}(\hat{\mathbf{p}}) +$ correction bias with respect to $\mathbf{f}_{\bar{S}}(\hat{\mathbf{p}})$.
7: **end procedure**

---

In the above general framework, the choice of $F$ depends on the symmetric property of interest. Later, in the case of entropy and distance to uniformity, we will choose $F$ to be the region where the empirical estimate fails; it is also the region that is difficult to estimate. One of the important properties of the above general framework is that $\mathbf{f}_S(\mathbf{p}^{\beta}_{\phi_S})$ (recall $\mathbf{p}^{\beta}_{\phi_S}$ is a $(\beta, S)$-approximate pseudo PML distribution and $\mathbf{f}_S(\mathbf{p}^{\beta}_{\phi_S})$ is the property value of distribution $\mathbf{p}^{\beta}_{\phi_S}$ on subset of domain elements $S \subseteq \mathcal{D}$) is close to $\mathbf{f}_S(\mathbf{p})$ with high probability. Below we state this result formally.

**Theorem 3.8.** *For any symmetric property* $\boldsymbol{f}$, *let* $G \subseteq \mathcal{D}$ *and* $F, F' \in 2^{\mathbb{Z}+}$. *If for all* $S' \in 2^G$, *there exists an estimator* $\hat{\boldsymbol{f}} : \Phi^n_{S'} \to \mathbb{R}$, *such that for any* $\boldsymbol{p}$ *and* $\phi_{S'} \sim \boldsymbol{p}$ *satisfies,*

$$\mathbb{P}\left(|\boldsymbol{f}_{S'}(\boldsymbol{p}) - \hat{\boldsymbol{f}}(\phi_{S'})| \geq 2\epsilon\right) \leq \delta \text{ and } \mathbb{P}\left(\text{Freq}\left(\phi_{S'}\right) \not\subseteq F'\right) \leq \gamma . \tag{3}$$

*Then for any sequence* $x^{2n} = (x_1^n, x_2^n)$,

$$\mathbb{P}\left(|\boldsymbol{f}_S(\boldsymbol{p}) - \boldsymbol{f}_S(\boldsymbol{p}^{\beta}_{\phi_S})| > 4\epsilon\right) \leq \frac{\delta n^{|F'|}}{\beta} + \gamma + \mathbb{P}\left(S \notin 2^G\right) ,$$

*where* $S$ *is a random set* $S \overset{\text{def}}{=} \{y \in \mathcal{D} \mid f(x_1^n, y) \in F\}$ *and* $\phi_S \overset{\text{def}}{=} \Phi_S(x_2^n)$.

Using the theorem above, we already have a good estimate for $\mathbf{f}_S(\mathbf{p})$ for appropriately chosen frequency subsets $F, F'$ and $G \subseteq \mathcal{D}$. Further, we choose these subsets $F, F'$ and $G$ carefully so that the empirical estimate $\mathbf{f}_{\bar{S}}(\hat{p})$ plus the correction bias with respect to $\mathbf{f}_{\bar{S}}$ is close to $\mathbf{f}_{\bar{S}}(\mathbf{p})$. Combining these together, we get the following results for entropy and distance to uniformity.

**Theorem 3.9.** *If error parameter* $\epsilon > \Omega\left(\frac{\log N}{N^{1-\alpha}}\right)$ *for any constant* $\alpha > 0$, *then for estimating entropy, the estimator 1 for* $\beta = n^{-\log n}$ *is sample complexity optimal.*

For entropy, we already know from [WY16a] that the empirical distribution is sample complexity optimal if $\epsilon < c\frac{\log N}{N}$ for some constant $c > 0$. Therefore the interesting regime for entropy estimation is when $\epsilon > \Omega\left(\frac{\log N}{N}\right)$ and our estimator works for almost all such $\epsilon$.

**Theorem 3.10.** *Let* $\alpha > 0$ *and error parameter* $\epsilon > \Omega\left(\frac{1}{N^{1-8\alpha}}\right)$, *then for estimating distance from uniformity, the estimator 1 for* $\beta = n^{-\sqrt{\frac{n \log n}{N}}}$ *is sample complexity optimal.*

Note that the estimator in [JHW17] also requires that the error parameter $\epsilon \geq \frac{1}{N^C}$, where $C > 0$ is some constant.

# 4 Analysis of General Framework for Symmetric Property Estimation

Here we provide proofs of the main results for our general framework (Theorem 3.7 and 3.8). These results weakly depend on the property and generalize results in [ADOS16]. The PML based estimator in [ADOS16] is sample competitive only for a restricted error parameter regime and this stems from the large number of possible profiles of length $n$. Our next lemma will be useful to address this issue and later we show how to use this result to prove Theorems 3.7 and 3.8.

**Lemma 4.1.** *For any subset* $S \subseteq \mathcal{D}$ *and* $\mathrm{F} \in 2^{\mathbb{Z}_+}$, *if set* $B$ *is defined as* $B \stackrel{\text{def}}{=} \{\phi_S \in \Phi_S^n \mid \mathrm{Freq}\,(\phi_S) \subseteq \mathrm{F}\}$, *then the cardinality of set* $B$ *is upper bounded by* $(n+1)^{|\mathrm{F}|}$.

*Proof of Theorem 3.7.* Using the law of total probability we have,

$$\mathbb{P}\left(|\mathbf{f}_S(\mathbf{p}) - \mathbf{f}_S(\mathbf{p}_{\phi_S}^\beta)| \geq 2\epsilon\right) = \mathbb{P}\left(|\mathbf{f}_S(\mathbf{p}) - \mathbf{f}_S(\mathbf{p}_{\phi_S}^\beta)| \geq 2\epsilon, \ \mathrm{Freq}\,(\phi_S) \subseteq \mathrm{F}\right)$$
$$+ \mathbb{P}\left(|\mathbf{f}_S(\mathbf{p}) - \mathbf{f}_S(\mathbf{p}_{\phi_S}^\beta)| \geq 2\epsilon, \ \mathrm{Freq}\,(\phi_S) \nsubseteq \mathrm{F}\right),$$
$$\leq \mathbb{P}\left(|\mathbf{f}_S(\mathbf{p}) - \mathbf{f}_S(\mathbf{p}_{\phi_S}^\beta)| \geq 2\epsilon, \ \mathrm{Freq}\,(\phi_S) \subseteq \mathrm{F}\right) + \mathbb{P}\left(\mathrm{Freq}\,(\phi_S) \nsubseteq \mathrm{F}\right).$$

Consider any $\phi_S \sim \mathbf{p}$. If $\mathbf{p}\,(\phi_S) > \delta/\beta$, then we know that $\mathbf{p}_{\phi_S}^\beta\,(\phi_S) > \delta$. For $\beta \leq 1$, we have $\mathbf{p}\,(\phi_S) > \delta$ that implies $|\mathbf{f}_S(\mathbf{p}) - \hat{\mathbf{f}}(\phi_S)| \leq \epsilon$. Further $\mathbf{p}_{\phi_S}^\beta\,(\phi_S) > \delta$ implies $|\mathbf{f}_S(\mathbf{p}_{\phi_S}^\beta) - \hat{\mathbf{f}}(\phi_S)| \leq \epsilon$. Using triangle inequality we get, $|\mathbf{f}_S(\mathbf{p}) - \mathbf{f}_S(\mathbf{p}_{\phi_S}^\beta)| \leq |\mathbf{f}_S(\mathbf{p}) - \hat{\mathbf{f}}(\phi_S)| + |\mathbf{f}_S(\mathbf{p}_{\phi_S}^\beta) - \hat{\mathbf{f}}(\phi_S)| \leq 2\epsilon$. Note we wish to upper bound the probability of set: $\mathbf{B}_{\mathrm{F},S,\hat{\mathbf{f}}} \stackrel{\text{def}}{=} \{\phi_S \in \Phi_S^n \mid \mathrm{Freq}\,(\phi_S) \subseteq \mathrm{F} \text{ and } |\mathbf{f}_S(\mathbf{p}) - \mathbf{f}_S(\mathbf{p}_{\phi_S}^\beta)| \geq 2\epsilon\}$. From the previous discussion, we get $\mathbf{p}\,(\phi_S) \leq \delta/\beta$ for all $\phi_S \in \mathbf{B}_{\mathrm{F},S,\hat{\mathbf{f}}}$. Therefore,

$$\mathbb{P}\left(|\mathbf{f}_S(\mathbf{p}) - \mathbf{f}_S(\mathbf{p}_{\phi_S}^\beta)| \geq 2\epsilon, \ \mathrm{Freq}\,(\phi_S) \subseteq \mathrm{F}\right) = \sum_{\phi_S \in \mathbf{B}_{\mathrm{F},S,\hat{\mathbf{f}}}} \mathbf{p}\,(\phi_S) \leq \frac{\delta}{\beta}|\mathbf{B}_{\mathrm{F},S,\hat{\mathbf{f}}}| \leq \frac{\delta}{\beta}(n+1)^{|\mathrm{F}|}.$$

In the final inequality, we use $\mathbf{B}_{\mathrm{F},S,\hat{\mathbf{f}}} \subseteq \{\phi_S \in \Phi_S^n \mid \mathrm{Freq}\,(\phi_S) \subseteq \mathrm{F}\}$ and invoke Lemma 4.1. □

*Proof for Theorem 3.8.* Using Bayes rule we have:

$$\mathbb{P}\left(|\mathbf{f}_S(\mathbf{p}) - \mathbf{f}_S(\mathbf{p}_{\phi_S}^\beta)| > 2\epsilon\right) = \sum_{S' \subseteq \mathcal{D}} \mathbb{P}\left(|\mathbf{f}_S(\mathbf{p}) - \mathbf{f}_S(\mathbf{p}_{\phi_S}^\beta)| > 2\epsilon \mid S = S'\right)\mathbb{P}\,(S = S')$$
$$\leq \sum_{S' \in 2^{\mathrm{G}}} \mathbb{P}\left(|\mathbf{f}_S(\mathbf{p}) - \mathbf{f}_S(\mathbf{p}_{\phi_S}^\beta)| > 2\epsilon \mid S = S'\right)\mathbb{P}\,(S = S') + \mathbb{P}\left(S \notin 2^{\mathrm{G}}\right).$$

(4)

In the second inequality, we use $\sum_{S' \notin 2^{\mathrm{G}}} \mathbb{P}\left(|\mathbf{f}_S(\mathbf{p}) - \mathbf{f}_S(\mathbf{p}_{\phi_S}^\beta)| > 2\epsilon, \ S = S'\right) \leq \mathbb{P}\left(S \notin 2^{\mathrm{G}}\right)$. Consider the first term on the right side of the above expression and note that it is upper bounded by, $\sum_{S' \in 2^{\mathrm{G}}} \mathbb{P}\left(|\mathbf{f}_{S'}(\mathbf{p}) - \mathbf{f}_{S'}(\mathbf{p}_{\phi_{S'}}^\beta)| > 2\epsilon\right)\mathbb{P}\,(S = S') \leq \sum_{S' \in 2^{\mathrm{G}}} \left[\frac{\delta n^{|F'|}}{\beta} + \mathbb{P}\,(\mathrm{Freq}\,(\phi_{S'}) \nsubseteq \mathrm{F}')\right]\mathbb{P}\,(S = S') \leq \frac{\delta n^{|F'|}}{\beta} + \gamma$. In the first upper bound, we removed randomness associated with the random set $S$ and used $\mathbb{P}\left(|\mathbf{f}_S(\mathbf{p}) - \mathbf{f}_S(\mathbf{p}_{\phi_S}^\beta)| > 2\epsilon \mid S = S'\right) = \mathbb{P}\left(|\mathbf{f}_{S'}(\mathbf{p}) - \mathbf{f}_{S'}(\mathbf{p}_{\phi_{S'}}^\beta)| > 2\epsilon\right)$. In the first inequality above, we invoke Theorem 3.7 using conditions from Equation (3). In the second inequality, we use $\sum_{S' \in 2^{\mathrm{G}}} \mathbb{P}\,(S = S') \leq 1$ and $\mathbb{P}\,(\mathrm{Freq}\,(\phi_S) \nsubseteq \mathrm{F}', S = S') \leq \gamma$. The theorem follows by combining all the analysis together. □

# 5 Applications of the General Framework

Here we provide applications of our general framework (defined in Section 3) using results from the previous section. We apply our general framework to estimate entropy and distance to uniformity. In Section 5.1 and Section 5.2 we analyze the performance of our estimator for entropy and distance to uniformity estimation respectively.

## 5.1 Entropy estimation

In order to prove our main result for entropy (Theorem 3.9), we first need the existence of an estimator for entropy with some desired properties. The existence of such an estimator will be crucial to bound the failure probability of our estimator. A result analogous to this is already known in [ADOS16] (Lemma 2) and the proof of our result follows from a careful observation of [ADOS16, WY16a]. We state this result here but defer the proof to appendix.

**Lemma 5.1.** *Let $\alpha > 0$, $\epsilon > \Omega\left(\frac{\log N}{N^{1-\alpha}}\right)$ and $S \subseteq \mathcal{D}$, then for entropy on subset $S$ ($\sum_{y \in S} \boldsymbol{p}_y \log \frac{1}{\boldsymbol{p}_y}$) there exists an $S$-pseudo profile based estimator that use the optimal number of samples, has bias less than $\epsilon$ and if we change any sample, changes by at most $c \cdot \frac{n^\alpha}{n}$, where $c$ is a constant.*

Combining the above lemma with Theorem 3.8, we next prove that our estimator defined in Algorithm 1 is sample complexity optimal for estimating entropy in a broader regime of error $\epsilon$.

*Proof for Theorem 3.9.* Let $\mathbf{f}(\mathbf{p})$ represent the entropy of distribution $\mathbf{p}$ and $\hat{\mathbf{f}}$ be the estimator in Lemma 5.1. Define $\mathrm{F} \overset{\text{def}}{=} [0, c_1 \log n]$ for constant $c_1 \geq 40$. Given the sequence $x^{2n}$, the random set $S$ is defined as $S \overset{\text{def}}{=} \{y \in \mathcal{D} \mid f(x_1^n, y) \leq c_1 \log n\}$. Let $\mathrm{F}' \overset{\text{def}}{=} [0, 8c_1 \log n]$, then by derivation in Lemma 6 [ADOS16] (or by simple application of Chernoff [4]) we have,

$$\mathbb{P}\left(\mathrm{Freq}\left(\phi_S\right) \not\subseteq \mathrm{F}'\right) = \mathbb{P}\left(\exists y \in \mathcal{D} \text{ such that } \mathbf{f}(x_1^n, y) \leq c_1 \log n \text{ and } \mathbf{f}(x_2^n, y) > 8c_1 \log n\right) \leq \frac{1}{n^5}.$$

Further let $\mathrm{G} \overset{\text{def}}{=} \{x \in \mathcal{D} \mid \mathbf{p}_x \leq \frac{2c_1 \log N}{n}\}$, then by Equation 48 in [WY16a] we have, $\mathbb{P}\left(S \notin 2^\mathrm{G}\right) \leq \frac{1}{n^4}$. Further for all $S' \in 2^\mathrm{G}$ we have,

$$\mathbb{P}\left(\mathrm{Freq}\left(\phi_{S'}\right) \not\subseteq \mathrm{F}'\right) = \mathbb{P}\left(\exists y \in S' \text{ such that } \mathbf{f}(x_2^n, y) > 8c_1 \log n\right) \leq \gamma \text{ for } \gamma = \frac{1}{n^5}.$$

Note for all $x \in S'$, $\mathbf{p}_x \leq \frac{2c_1 \log N}{n}$ and the above inequality also follows from Chernoff. All that remains now is to upper bound $\delta$. Using the estimator constructed in Lemma 5.1 and further combined with McDiarmid's inequality, we have,

$$\mathbb{P}\left(|\mathbf{f}_{S'}(\mathbf{p}) - \hat{\mathbf{f}}(\phi_{S'})| \geq 2\epsilon\right) \leq 2 \exp\left(\frac{-2\epsilon^2}{n(c\frac{n^\alpha}{n})^2}\right) \leq \delta \text{ for } \delta = \exp\left(-2\epsilon^2 n^{1-2\alpha}\right).$$

Substituting all these parameters together in Theorem 3.8 we have,

$$\mathbb{P}\left(|\mathbf{f}_S(\mathbf{p}) - \mathbf{f}_S(\mathbf{p}_{\phi_S}^\beta)| > 2\epsilon\right) \leq \frac{\delta n^{|F'|}}{\beta} + \mathbb{P}\left(\mathrm{Freq}\left(\phi_S\right) \not\subseteq \mathrm{F}'\right) + \mathbb{P}\left(S \notin 2^\mathrm{G}\right)$$
$$\leq \exp\left(-2\epsilon^2 n^{1-2\alpha}\right) n^{9c_1 \log n} + \frac{1}{n^4} \leq \frac{2}{n^4}. \tag{5}$$

In the first inequality, we use Theorem 3.8. In the second inequality, we substituted the values for $\delta, \gamma, \beta$ and $\mathbb{P}\left(S \notin 2^\mathrm{G}\right)$. In the final inequality we used $n = \Theta(\frac{N}{\log N} \frac{1}{\epsilon})$ and $\epsilon > \Omega\left(\frac{\log^3 N}{N^{1-4\alpha}}\right)$.

Our final goal is to estimate $\mathbf{f}(\mathbf{p})$, and to complete the proof we need to argue that $\mathbf{f}_{\bar{S}}(\hat{\mathbf{p}})$ + the correction bias with respect to $\mathbf{f}_{\bar{S}}$ is close to $\mathbf{f}_{\bar{S}}(\mathbf{p})$, where recall $\hat{\mathbf{p}}$ is the empirical distribution on sequence $x_2^n$. The proof for this follows immediately from [WY16a] (Case 2 in the proof of Proposition 4). [WY16a] bound the bias and variance of the empirical estimator with a correction bias and applying Markov inequality on their result we get $\mathbb{P}\left(|\mathbf{f}_{\bar{S}}(\mathbf{p}) - (\mathbf{f}_{\bar{S}}(\hat{\mathbf{p}}) + \frac{|\bar{S}|}{n})| > 2\epsilon\right) \leq \frac{1}{3}$, where $\frac{|\bar{S}|}{n}$ is the correction bias in [WY16a]. Using triangle inequality, our estimator fails if either $|\mathbf{f}_{\bar{S}}(\mathbf{p}) - (\mathbf{f}_{\bar{S}}(\hat{\mathbf{p}}) + \frac{|\bar{S}|}{n})| > 2\epsilon$ or $|\mathbf{f}_S(\mathbf{p}) - \mathbf{f}_S(\mathbf{p}_{\phi_S}^\beta)| > 2\epsilon$. Further by union bound the failure probability is at most $\frac{1}{3} + \frac{2}{n^4}$, which is a constant. $\square$

## 5.2 Distance to Uniformity estimation

Here we prove our main result for distance to uniformity estimation (Theorem 3.10). First, we show existence of an estimator for distance to uniformity with certain desired properties. Similar to entropy, a result analogous to this is shown in [ADOS16] (Lemma 2) and the proof of our result follows from the careful observation of [ADOS16, JHW17]. We state this result here but defer the proof to Appendix C.

**Lemma 5.2.** *Let $\alpha > 0$ and $S \subseteq \mathcal{D}$, then for distance to uniformity on $S$ ($\sum_{y \in S} |\boldsymbol{p}_y - \frac{1}{N}|$) there exists an $S$-pseudo profile based estimator that use the optimal number of samples, has bias at most $\epsilon$ and if we change any sample, changes by at most $c \cdot \frac{n^\alpha}{n}$, where $c$ is a constant.*

Combining the above lemma with Theorem 3.8 we provide the proof for Theorem 3.10.

*Proof for Theorem 3.10.* Let $\mathbf{f}(\mathbf{p})$ represent the distance to uniformity for distribution $\mathbf{p}$ and $\hat{\mathbf{f}}$ be the estimator in Lemma 5.2. Define $F = [\frac{n}{N} - \sqrt{\frac{c_1 n \log n}{N}}, \frac{n}{N} + \sqrt{\frac{c_1 n \log n}{N}}]$ for some constant $c_1 \geq 40$. Given the sequence $x^{2n}$, the random set $S$ is defined as $S \stackrel{\text{def}}{=} \{y \in \mathcal{D} \mid f(x_1^n, y) \in F\}$. Let $F' = [\frac{n}{N} - \sqrt{\frac{8c_1 n \log n}{N}}, \frac{n}{N} + \sqrt{\frac{8c_1 n \log n}{N}}]$, then by derivation in Lemma 7 of [ADOS16] (also shown in [JHW17] [5]) we have,

$$\mathbb{P}\left(\text{Freq}\,(\phi_S) \not\subseteq F'\right) = \mathbb{P}\left(\exists y \in \mathcal{D} \text{ such that } \mathbf{f}(x_1^n, y) \in F \text{ and } \mathbf{f}(x_2^n, y) \notin F'\right) \leq \frac{1}{n^4} \,.$$

Further let $G \stackrel{\text{def}}{=} \{x \in \mathcal{D} \mid \mathbf{p}_x \in [\frac{1}{N} - \sqrt{\frac{2c_1 \log n}{nN}}, \frac{1}{N} + \sqrt{\frac{2c_1 \log n}{nN}}]\}$, then using Lemma 2 in [JHW17] we get,

$$\mathbb{P}\left(S \notin 2^G\right) = \mathbb{P}\left(\exists y \in \mathcal{D} \text{ such that } \mathbf{f}(x_1^n, y) \in F \text{ and } \mathbf{p}_x \notin G\right) \leq \frac{\log n}{n^{1-\epsilon}} \,.$$

Further for all $S' \in 2^G$ we have,

$$\mathbb{P}\left(\text{Freq}\,(\phi_{S'}) \not\subseteq F'\right) = \mathbb{P}\left(\exists y \in S' \text{ such that } \mathbf{f}(x_2^n, y) > 8c_1 \log n\right) \leq \gamma \text{ for } \gamma = \frac{1}{n} \,.$$

Note for all $x \in S'$, $\mathbf{p}_x \in G$ and the above result follows from [JHW17] (Lemma 1). All that remains now is to upper bound $\delta$. Using the estimator constructed in Lemma 5.2 and further combined with McDiarmid's inequality, we have,

$$\mathbb{P}\left(|\mathbf{f}_{S'}(\mathbf{p}) - \hat{\mathbf{f}}(\phi_{S'})| \geq 2\epsilon\right) \leq 2\exp\left(\frac{-2\epsilon^2}{n(c\frac{n^\alpha}{n})^2}\right) \leq \delta \text{ for } \delta = \exp\left(-2\epsilon^2 n^{1-2\alpha}\right) \,.$$

Substituting all these parameters in Theorem 3.8 we get,

$$\mathbb{P}\left(|\mathbf{f}_S(\mathbf{p}) - \mathbf{f}_S(\mathbf{p}_{\phi_S}^\beta)| > 2\epsilon\right) \leq \frac{\delta n^{|F'|}}{\beta} + \mathbb{P}\left(\text{Freq}\,(\phi_S) \not\subseteq F'\right) + \mathbb{P}\left(S \notin 2^G\right) \tag{6}$$

$$\leq \exp\left(-2\epsilon^2 n^{1-2\alpha}\right) n^{2\sqrt{\frac{8c_1 n \log n}{N}}} + \frac{\log n}{n^{1-\epsilon}} + \frac{1}{n} \leq o(1) \,.$$

In the first inequality, we use Theorem 3.8. In the second inequality, we substituted values for $\delta, \gamma, \beta$ and $\mathbb{P}\left(S \notin 2^G\right)$. In the final inequality we used $n = \Theta(\frac{N}{\log N} \frac{1}{\epsilon^2})$ and $\epsilon > \Omega\left(\frac{1}{N^{1-8\alpha}}\right)$.

Our final goal is to estimate $\mathbf{f}(\mathbf{p})$, and to complete the proof we argue that $\mathbf{f}_{\bar{S}}(\hat{\mathbf{p}})$ + correction bias with respect to $\mathbf{f}_{\bar{S}}$ is close to $\mathbf{f}_{\bar{S}}(\mathbf{p})$, where recall $\hat{\mathbf{p}}$ is the empirical distribution on sequence $x_2^n$. The proof for this case follows immediately from [JHW17] (proof of Theorem 2). [JHW17] define three kinds of events $\mathcal{E}_1, \mathcal{E}_2$ and $\mathcal{E}_3$, the proof for our empirical case follows from the analysis of bias and variance of events $\mathcal{E}_1$ and $\mathcal{E}_2$. Further combining results in [JHW17] with Markov inequality we get $\mathbb{P}(|\mathbf{f}_{\bar{S}}(\mathbf{p}) - \mathbf{f}_{\bar{S}}(\hat{\mathbf{p}})| > 2\epsilon) \leq \frac{1}{3}$, and the correction bias here is zero. Using triangle inequality, our estimator fails if either $|\mathbf{f}_{\bar{S}}(\mathbf{p}) - (\mathbf{f}_{\bar{S}}(\hat{\mathbf{p}}) + \frac{|\bar{S}|}{n})| > 2\epsilon$ or $|\mathbf{f}_S(\mathbf{p}) - \mathbf{f}_S(\mathbf{p}_{\phi_S}^\beta)| > 2\epsilon$. Further by union bound the failure probability is upper bounded by $\frac{1}{3} + o(1)$, which is a constant. $\square$

# 6 Experiments

We performed two different sets of experiments for entropy estimation – one to compare performance guarantees and the other to compare running times. In our pseudo PML approach, we divide the samples into two parts. We run the empirical estimate on one (this is easy) and the PML estimate on the other. For the PML estimate, any algorithm to compute an approximate PML distribution can be used in a black box fashion. An advantage of the pseudo PML approach is that it can use any algorithm to estimate the PML distribution as a black box, providing both competitive performance and running time efficiency. In our experiments, we use the heuristic algorithm in [PJW17] to compute an approximate PML distribution. In the first set of experiments detailed below, we compare the performance of the pseudo PML approach with raw [PJW17] and other state-of-the-art estimators for estimating entropy. Our code is available at https://github.com/shiragur/CodeForPseudoPML.git

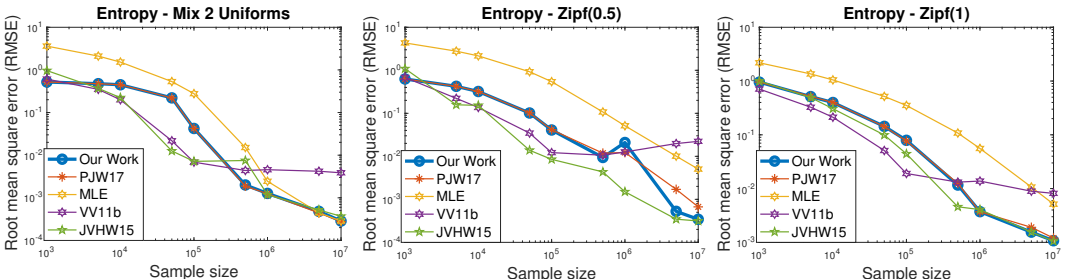

Each plot depicts the performance of various algorithms for estimating entropy of different distributions with domain size $N = 10^5$. Each data point represents 50 random trials. "Mix 2 Uniforms" is a mixture of two uniform distributions, with half the probability mass on the first $N/10$ symbols, and $\text{Zipf}(\alpha) \sim 1/i^\alpha$ with $i \in [N]$. MLE is the naive approach of using the empirical distribution with correction bias; all the remaining algorithms are denoted using bibliographic citations. In our algorithm we pick $threshold = 18$ (same as [WY16a]) and our set F $= [0, 18]$ (input of Algorithm 1), i.e. we use the PML estimate on frequencies $\leq 18$ and empirical estimate on the rest. Unlike Algorithm 1, we do not perform sample splitting in the experiments – we believe this requirement is an artifact of our analysis. For estimating entropy, the error achieved by our estimator is competitive with [PJW17] and other state-of-the-art entropy estimators. Note that our results match [PJW17] for small sample sizes because not many domain elements cross the threshold and for a large fraction of the samples, we simply run the [PJW17] algorithm.

In the second set of experiments we demonstrate the running time efficiency of our approach. In these experiments, we compare the running time of our algorithm using [PJW17] as a subroutine to the raw [PJW17] algorithm on the $\text{Zipf}(1)$ distribution. The second row is the fraction of samples on which our algorithm uses the empirical estimate (plus correction bias). The third row is the ratio of the running time of [PJW17] to our algorithm. For large sample sizes, the entries in the EmpFrac row have high value, i.e. our algorithm applies the simple empirical estimate on large fraction of samples; therefore, enabling 10x speedup in the running times.

| Samples size | $10^3$ | $5 * 10^3$ | $10^4$ | $5 * 10^4$ | $10^5$ | $5 * 10^5$ | $10^6$ | $5 * 10^6$ |
|---|---|---|---|---|---|---|---|---|
| EmpFrac | 0.184 | 0.317 | 0.372 | 0.505 | 0.562 | 0.695 | 0.752 | 0.886 |
| Speedup | 0.824 | 1.205 | 1.669 | 3.561 | 4.852 | 9.552 | 13.337 | 12.196 |

## Acknowledgments

We thank the reviewers for the helpful comments, great suggestions, and positive feedback. Moses Charikar was supported by a Simons Investigator Award, a Google Faculty Research Award and an Amazon Research Award. Aaron Sidford was partially supported by NSF CAREER Award CCF-1844855.

## Footnotes

[1]Throughout the paper, distribution refers to discrete distribution.

[2]The profile does not contain $\phi(0)$, the number of unseen domain elements.

[3] Under the constraint that its minimum non-zero probability value is at least $\frac{1}{k}$. This assumption is also necessary for the results in [ADOS16] to hold.

[4]Note probability of many events in this proof can be easily bounded by application of Chernoff. These bounds on probabilities are also shown in [ADOS16, WY16a] and we use these inequalities by omitting details.

[5]Similar to entropy, for many events their probabilities can be bounded by simple application of Chernoff and have already been shown in [ADOS16, JHW17]. We omit details for these inequalities.

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
