[Supplementary Material]

# A  Support Estimation

Here we study the PML based plug-in estimator for support estimation. [ADOS16] showed that PML based plug-in estimator is sample complexity optimal for estimating support within additive accuracy $\epsilon k$ for all $\epsilon > \frac{1}{k^{0.2499}}$. Further for any $\epsilon < \frac{1}{k^\delta}$ for some constant $\delta > 0$, the empirical distribution based plug-in estimator is exact with high probability. Here we provide proofs for two main results described in Section 3 for support. In Theorem 3.1 and Theorem 3.2, we show that PML and approximate PML distributions (under the constraint that all its probability values are $\geq \frac{1}{k}$) based plug-in estimators are sample complexity optimal for all parameter regimes, thus providing a better analysis for [ADOS16].

We next define a function that outputs the number of distinct frequencies in the profile. Later in Lemma A.2, we show that the support of PML and approximate PML distribution is at least the number of distinct elements in the sequence.

**Definition A.1.** For any $S \subseteq \mathcal{D}$, the function Distinct : $\Phi^n \to \mathbb{Z}_+$, takes input $\phi$ and returns $\sum_{j \in [n]} \phi_j$. For any sequence $x^n$, we overload notation and use Distinct($x^n$) to denote Distinct($\Phi(x^n)$). Note Distinct($\phi$) and Distinct($x^n$) denote the number of distinct domain elements observed in profile $\phi$ or sequence $x^n$ respectively.

**Lemma A.2.** *For any distribution $\boldsymbol{p} \in \Delta^{\mathcal{D}}$ such that $\boldsymbol{p}_x \in \{0\} \cup [\frac{1}{k}, 1]$ and a profile $\phi \in \Phi^n$, if $S(\boldsymbol{p}) < \text{Distinct}(\phi)$ then $\boldsymbol{p}(\phi) = 0$.*

*Proof.* Consider sequences $x^n$ with $\Phi(x^n) = \phi$. All such sequences have Distinct($\phi$) number of distinct observed elements that is strictly greater than $S(\mathbf{p})$ and distribution $\mathbf{p}$ assigns probability zero for all these sequences. $\qquad \square$

*Proof for Theorem 3.1.* Given $\phi$, let $\mathbf{p}_\phi \in \Delta^{\mathcal{D}}$, be the distribution with $\mathbf{p}_\phi(x) \in \{0\} \cup [\frac{1}{k}, 1]$. If $\epsilon > \frac{1}{k^{0.2499}}$, we know that plug-in approach on $\mathbf{p}_\phi$ is sample complexity optimal [ADOS16]. We consider the regime where $\epsilon \leq \frac{1}{k^{0.2499}}$ and here the number of samples $n = c \cdot k \log k$ for some constant $c \geq 2$. If $S(\mathbf{p}_\phi) < \text{Distinct}(\phi)$, then by Lemma A.2 we have $\mathbf{p}_\phi(\phi) = 0$ a contradiction because the empirical distribution assigns a non-zero probability value for observing $\phi$. Therefore, without loss of generality we assume $S(\mathbf{p}_\phi) \geq \text{Distinct}(\phi)$. We next argue that $S(\mathbf{p}_\phi) = \text{Distinct}(\phi)$. We prove this statement by contradiction. Suppose $S(\mathbf{p}_\phi) > \text{Distinct}(\phi)$, then define $\mathbf{p}'_\phi \in \Delta^{\mathcal{D}}$ to be the PML distribution under constraints $S(\mathbf{p}'_\phi) = \text{Distinct}(\phi)$ and $\mathbf{p}'_\phi(x) \in \{0\} \cup [\frac{1}{k}, 1]$. Let Support : $\Delta^{\mathcal{D}} \to 2^{\mathcal{D}}$ be a function that takes distribution $\mathbf{p}$ as input and returns index set for the support of $\mathbf{p}$. Now consider $\mathbb{P}(\mathbf{p}_\phi, \phi)$, and recall $\mathbb{P}(\mathbf{p}_\phi, \phi) = \sum_{\{x^n \in \mathcal{D}^n \mid \Phi(x^n) = \phi\}} \mathbb{P}(\mathbf{p}_\phi, x^n)$. Further note that $\sum_{\{x^n \in \mathcal{D}^n \mid \Phi(x^n) = \phi\}} \mathbb{P}(\mathbf{p}_\phi, x^n) = \sum_{\{S \subseteq \text{Support}(\mathbf{p}_\phi) \mid |S| = \text{Distinct}(\phi)\}} \sum_{\{x^n \in S^n \mid \Phi(x^n) = \phi\}} \mathbb{P}(\mathbf{p}_\phi, x^n)$, therefore,

$$
\begin{aligned}
\mathbb{P}(\mathbf{p}_\phi, \phi) &= \sum_{\{S \subseteq \text{Support}(\mathbf{p}_\phi) \mid |S| = \text{Distinct}(\phi)\}} \sum_{\{x^n \in S^n \mid \Phi(x^n) = \phi\}} \mathbb{P}(\mathbf{p}_\phi, x^n) \\
&\leq \sum_{\{S \subseteq \text{Support}(\mathbf{p}_\phi) \mid |S| = \text{Distinct}(\phi)\}} \left(1 - \frac{S(\mathbf{p}_\phi) - |S|}{k}\right)^n \mathbb{P}(\mathbf{p}'_\phi, \phi) .
\end{aligned}
\tag{7}
$$

In the second inequality, we use for all $x \in \mathcal{D}$, $\mathbf{p}_\phi(x) \in \{0\} \cup [\frac{1}{k}, 1]$ and we have $\sum_{x \in S} \mathbf{p}_\phi(x) \leq \left(1 - \frac{S(\mathbf{p}_\phi) - |S|}{k}\right)$ and the inequality follows. We next upper bound the term $\sum_{\{S \subseteq \text{Support}(\mathbf{p}_\phi) \mid |S| = \text{Distinct}(\phi)\}} \left(1 - \frac{S(\mathbf{p}_\phi) - |S|}{k}\right)^n$. Note that, $\sum_{\{S \subseteq \text{Support}(\mathbf{p}_\phi) \mid |S| = \text{Distinct}(\phi)\}} \left(1 - \frac{S(\mathbf{p}_\phi) - |S|}{k}\right)^n \leq \exp\left(-n \frac{S(\mathbf{p}_\phi) - \text{Distinct}(\phi)}{k}\right) \binom{S(\mathbf{p}_\phi)}{\text{Distinct}(\phi)} \leq \exp\left(-n \frac{S(\mathbf{p}_\phi) - \text{Distinct}(\phi)}{k} + (S(\mathbf{p}_\phi) - \text{Distinct}(\phi)) \log S(\mathbf{p}_\phi)\right) \leq \exp(-\log k)$. In the second inequality, we use a weak upper bound on the quantity $\binom{S(\mathbf{p}_\phi)}{\text{Distinct}(\phi)}$. In the third and fourth inequality, we use $n = ck \log k$, $c \geq 2$ and $k \geq S(\mathbf{p}_\phi) > \text{Distinct}(\phi)$. Combining everything together we get, $\mathbb{P}(\mathbf{p}_\phi, \phi) \leq \exp(-\log k) \mathbb{P}(\mathbf{p}'_\phi, \phi)$. A contradiction because $\mathbf{p}_\phi$ is the PML distribution.

Therefore if $n > 2k \log k$, then the previous derivation implies,

$$\mathbb{P}\left(S(\mathbf{p}_\phi) = \text{Distinct}(\phi)\right) = 1 .$$

Further if $n > 2k \log k$, then

$$\mathbb{P}\left(S(\mathbf{p}) = \text{Distinct}(\phi)\right) \geq 1 - k \exp\left(\frac{-n}{k}\right) .$$

Combining previous two inequalities and substituting $n > 2k \log k$ we get, $\mathbb{P}\left(S(\mathbf{p}) = S(\mathbf{p}_\phi)\right) \geq 1 - \exp(-\log k)$, thus concluding the proof. $\qquad\square$

*Proof for Theorem 3.2.* The proof for this result is similar to Theorem 3.1 and for completeness we reprove it. Given $\phi$, let $\mathbf{p}_\phi, \mathbf{p}_\phi^\beta \in \Delta^{\mathcal{D}}$, be PML and $\beta$-approximate PML distributions respectively under the constraint $\mathbf{p}_\phi(x), \mathbf{p}_\phi^\beta(x) \in \{0\} \cup [\frac{1}{k}, 1]$. If $\epsilon > \frac{1}{k^{0.2499}}$, by [ADOS16] we already know that plug-in approach on $\mathbf{p}_\phi^\beta$ for $\beta = \exp(-\epsilon^2 n^{1-\alpha})$ is sample complexity optimal with high probability. Here we consider the regime $\epsilon \leq \frac{1}{k^{0.2499}}$ and in this case the number of samples $n = c \cdot k \log k$ for some large constant $c \geq 2$. If $S(\mathbf{p}_\phi^\beta) < \text{Distinct}(\phi)$, then by Lemma A.2 we have $\mathbf{p}_\phi^\beta(\phi) = 0$ which is a contradiction, because the empirical distribution clearly returns a non-zero probability value. Therefore, without loss of generality we assume $S(\mathbf{p}_\phi^\beta) \geq \text{Distinct}(\phi)$. We next argue that $S(\mathbf{p}_\phi^\beta) \leq \text{Distinct}(\phi) + \epsilon k$. We prove this statement by contradiction. Suppose $S(\mathbf{p}_\phi^\beta) > \text{Distinct}(\phi) + \epsilon k$, then consider the $\mathbb{P}\left(\mathbf{p}_\phi^\beta, \phi\right)$, and recall $\mathbb{P}\left(\mathbf{p}_\phi^\beta, \phi\right) = \sum_{\{x^n \in \mathcal{D}^n | \Phi(x^n) = \phi\}} \mathbb{P}\left(\mathbf{p}_\phi^\beta, x^n\right)$. Further note that $\sum_{\{x^n \in \mathcal{D}^n | \Phi(x^n) = \phi\}} \mathbb{P}\left(\mathbf{p}_\phi^\beta, x^n\right) = \sum_{\{S \subseteq \text{Support}(\mathbf{p}_\phi^\beta) | |S| = \text{Distinct}(\phi)\}} \sum_{\{x^n \in S^n | \Phi(x^n) = \phi\}} \mathbb{P}\left(\mathbf{p}_\phi^\beta, x^n\right)$. Therefore,

$$
\begin{aligned}
\mathbb{P}\left(\mathbf{p}_\phi^\beta, \phi\right) &= \sum_{\{S \subseteq \text{Support}(\mathbf{p}_\phi^\beta) | |S| = \text{Distinct}(\phi)\}} \sum_{\{x^n \in S^n | \Phi(x^n) = \phi\}} \mathbb{P}\left(\mathbf{p}_\phi^\beta, x^n\right) \\
&\leq \sum_{\{S \subseteq \text{Support}(\mathbf{p}_\phi^\beta) \,|\, |S| = \text{Distinct}(\phi)\}} \left(1 - \frac{S(\mathbf{p}_\phi^\beta) - |S|}{k}\right)^n \mathbb{P}\left(\mathbf{p}_\phi, \phi\right) .
\end{aligned}
\tag{8}
$$

In the final inequality we used for all $x \in \mathcal{D}$, $\mathbf{p}_\phi^\beta(x) \in \{0\} \cup [\frac{1}{k}, 1]$ and we have $\sum_{x \in S} \mathbf{p}_\phi^\beta(x) \leq \left(1 - \frac{S(\mathbf{p}_\phi^\beta) - |S|}{k}\right)$ and using the definition of $\mathbf{p}_\phi$ the inequality follows. We next upper bound the term $\sum_{\{S \subseteq \text{Support}(\mathbf{p}_\phi^\beta) \,|\, |S| = \text{Distinct}(\phi)\}} \left(1 - \frac{S(\mathbf{p}_\phi^\beta) - |S|}{k}\right)^n$. Note that $\sum_{\{S \subseteq \text{Support}(\mathbf{p}_\phi^\beta) \,|\, |S| = \text{Distinct}(\phi)\}} (1 - \frac{S(\mathbf{p}_\phi^\beta) - |S|}{k})^n \leq \exp(-n \frac{S(\mathbf{p}_\phi^\beta) - \text{Distinct}(\phi)}{k}) \binom{S(\mathbf{p}_\phi^\beta)}{\text{Distinct}(\phi)} \leq \exp(-n \frac{S(\mathbf{p}_\phi^\beta) - \text{Distinct}(\phi)}{k} + (S(\mathbf{p}_\phi^\beta) - \text{Distinct}(\phi)) \log S(\mathbf{p}_\phi^\beta)) \leq \exp((S(\mathbf{p}_\phi^\beta) - \text{Distinct}(\phi))(\log k - c \log k) \leq \exp(-\epsilon k \log k) < \exp(-\epsilon^2 n^{1-4\alpha})$. In the second inequality, we use a weak upper bound for the quantity $\binom{S(\mathbf{p}_\phi)}{\text{Distinct}(\phi)}$. In the third and fourth inequality, we use $n = ck \log k$, $c \geq 2$ and $S(\mathbf{p}_\phi^\beta) > \text{Distinct}(\phi) + \epsilon k$. In the final inequality, we use $n^{1-\alpha} \leq k \log k$ for constant $\alpha > 0$. Combining everything together we get $\mathbb{P}\left(\mathbf{p}_\phi^\beta, \phi\right) < \exp(-\epsilon^2 n^{1-4\alpha}) \mathbb{P}\left(\mathbf{p}_\phi, \phi\right)$, a contradiction on the definition of $\mathbf{p}_\phi^\beta$.

Therefore if $n > 2k \log k$, then the previous derivation implies $\mathbb{P}\left(|S(\mathbf{p}_\phi^\beta) - \text{Distinct}(\phi)| \geq \epsilon k\right) = 1$. Further if $n > 2k \log k$, then $\mathbb{P}\left(S(\mathbf{p}) = \text{Distinct}(\phi)\right) \geq 1 - k \exp(\frac{-n}{k})$. Combining the previous two inequalities and substituting $n > 2k \log k$ we get, $\mathbb{P}\left(|S(\mathbf{p}_\phi^\beta) - S(\mathbf{p})| \geq \epsilon k\right) \geq 1 - \exp(-\log k)$, thus concluding the proof. $\qquad\square$

# B Omitted Proof from Section 4

Here we provide the proof for Lemma 4.1.

*Proof for Lemma 4.1.* Fix an ordering on the elements of F. Let $F(i)$ denote the $i$'th frequency element of F. For all $\phi_S \in B$, the set of distinct frequencies in $\phi_S$ is a subset of F and the length of $\phi_S$ is equal to $n$. Therefore, any element $\phi_S \in B$ can be encoded as a unique vector $v_{\phi_S} \in [0, n]^F$, where $v_{\phi_S}(i) \stackrel{\text{def}}{=} \phi(F(i))$ denotes the number of elements in $\phi_S$ that have frequency $F(i)$. Using the previous discussion, we have $|B| \leq |[0, n]^F| \leq (n+1)^{|F|}$. □

# C Omitted Proofs from Section 5

Here, we present and prove results related to the existence of an estimator for entropy and distance to uniformity on a fixed subset $S \subseteq \mathcal{D}$. Note the estimator we provide here is exactly same to the one presented in [ADOS16] but defined only on subset $S \subseteq \mathcal{D}$. All the results and proofs presented here are similar to the ones in [ADOS16] and for completeness and verification purposes we reprove (with slight modifications) these results. As in [ADOS16], we first provide a general definition of an estimator that works both for entropy and distance to uniformity. In Lemma C.1, we prove a result that captures the maximum change of this general estimator by changing one sample. In section C.1 and C.2, we provide proofs for entropy and distance to uniformity respectively.

Given $2n$ samples $x^{2n} = (x_1^n, x_2^n)$ from distribution $\mathbf{p}$. Let $n_y' \stackrel{\text{def}}{=} \mathbf{f}(x_1^n, y)$, and $n_y \stackrel{\text{def}}{=} \mathbf{f}(x_2^n, y)$ be the number of appearances of symbol $y$ in the first and second half respectively. We define the following estimator which is exactly the same as [ADOS16] but defined only on subset $S$. For all $x \in S$,

$$\hat{g}_S(x^{2n}) = \max \left\{ \min \left\{ \sum_{y \in S} g_y, f_{S,\max} \right\}, 0 \right\}.$$

where $f_{S,\max}$ is the maximum value of the property $f$ on subset $S$ and for all $y \in S$,

$$g_y = \begin{cases} G_{L,g}(n_y), & \text{for } n_y' < c_2 \log N, \text{ and } n_y < c_1 \log N, \\ 0, & \text{for } n_y' < c_2 \log k, \text{ and } n_y \geq c_1 \log N, \\ g\left(\frac{n_y}{n}\right) + g_n, & \text{for } n_y' \geq c_2 \log N, \end{cases}$$

where $g_n$ is the first order bias correction term for $g$, $G_{L,g}(n_y) = \sum_{i=1}^{L} b_i \left(\frac{n_y}{n}\right)^i$ is the unbiased estimator for $P_{L,g}(\mathbf{p}_y)$, the optimal uniform approximation of function $g$ by degree-$L$ polynomials on $[0; c_1 \frac{\log n}{n}]$.

**Lemma C.1.** *For any estimator $\hat{g}$ defined as above, changing any one of the sample changes the estimator by at most*

$$9 \max \left( e^{L^2/n} \max |b_i|, \frac{L_g}{n}, g\left(\frac{c_1 \log(n)}{n}\right), g_n \right),$$

*where $L_g = n \max_{i \in \mathbb{N}} |g(i/n) - g((i-1)/n)|$.*

*Proof.* Given $2n$ samples $x^{2n} = (x_1^n, x_2^n)$ from distribution $\mathbf{p}$. Recall the estimator for entropy and distance to uniform from [ADOS16],

$$\hat{g}(x^{2n}) = \max \left\{ \min \left\{ \sum_{y \in S} g_y, f_{\max} \right\}, 0 \right\}.$$

where $f_{\max}$ is the maximum value of the property $f$ and for all $y \in \mathcal{D}$,

$$g_y = \begin{cases} G_{L,g}(n_y), & \text{for } n_y' < c_2 \log N, \text{ and } n_y < c_1 \log N, \\ 0, & \text{for } n_y' < c_2 \log k, \text{ and } n_y \geq c_1 \log N, \\ g\left(\frac{n_y}{n}\right) + g_n, & \text{for } n_y' \geq c_2 \log N, \end{cases}$$

Now construct a new sequence from $x^n$ as follows: replace all symbols in $\bar{S}$ (appearing in $x^n$) by a unique symbol $y' \in \bar{S}$ and call this new sequence $z^n$. Now note our estimator is unaffected by this change, because it only depends on the occurrences of elements in $S$. The change in the value of estimator in [ADOS16] by changing one sample in $z^n$ is upper bounded by:

$$8 \max\left(e^{L^2/n} \max |b_i|, \frac{L_g}{n}, g\left(\frac{c_1 \log(n)}{n}\right), g_n\right), \tag{9}$$

The above result follows by Lemma 5 in [ADOS16]. We next study the change in the value of our estimator by changing one sample in $z^n$. Note this is equivalent to the change in the value of our estimator by changing one sample in $x^n$. The worst case change in the value of our estimator is when we take a symbol in $S$ (or $\bar{S}\backslash\{y'\}$) and replace it by a symbol in $\bar{S}\backslash\{y'\}$ (or $S$). In this case, by triangle inequality change in our estimator is upper bounded by $8 \max\left(e^{L^2/n} \max |b_i|, \frac{L_g}{n}, g\left(\frac{c_1 \log(n)}{n}\right), g_n\right) + G_{L,g}(1)$ that is further upper bounded by $9 \max\left(e^{L^2/n} \max |b_i|, \frac{L_g}{n}, g\left(\frac{c_1 \log(n)}{n}\right), g_n\right)$ and the result follows.

$\square$

## C.1 Entropy

Here we present proof sketch for the following: for entropy the estimator defined above has low bias and the value of the estimator does not change too much by change in one sample. This result is analogous to Lemma 6 in [ADOS16] and our proof for this lemma is very similar to [ADOS16] and for completeness sketch for the proof.

**Lemma C.2.** *Let $g_n = 1/(2n)$ and $\alpha > 0$. Suppose $c_1 = 2c_2$, and $c_2 > 35$, Further suppose that $n^3\left(\frac{16c_1}{\alpha^2} + \frac{1}{c_2}\right) > \log k \cdot \log n$. Then for all subset $S \subseteq \mathcal{D}$, there exists a polynomial approximation of $-y \log y$ with degree $L = 0.25\alpha \log n$, over $[0, c_1 \frac{\log N}{n}]$ such that $\max_i |b_i| \leq n^\alpha/n$ and the bias of the entropy estimator on subset $S$ ($\sum_{y \in S} \boldsymbol{p}_y \log \frac{1}{\boldsymbol{p}_y}$) is at most $O\left(\left(1 + \frac{1}{\alpha^2}\right)\frac{N}{n \log N} + \frac{\log N}{N^4}\right)$.*

*Proof.* We first upper bound the bias of our estimator. We consider three events here. $E_1 \stackrel{\text{def}}{=} \cap_{y \in S}\left\{n'_y \leq c_2 \log N, n'_y \leq c_1 \log N \implies \boldsymbol{p}_y \leq \frac{c_1 \log N}{n}\right\}$, $E_2 \stackrel{\text{def}}{=} \cap_{y \in S}\left\{n'_y > c_2 \log N \implies \boldsymbol{p}_y > \frac{c_3 \log N}{n}\right\}$ and $E_3 \stackrel{\text{def}}{=} \cap_{y \in S}\left\{n'_y \leq c_2 \log N \text{ and } n'_y > c_1 \log N\right\}$. By proof of Lemma 6 in [ADOS16] we have,

$$\mathbb{P}\left(E_3^c\right) \leq \frac{1}{n^{4.9}} \tag{10}$$

By equations 48 and 49 in [WY16a] combined with Equation (10), we get,

$$\mathbb{P}\left(E_1^c\right) \leq \frac{2}{N^4} \text{ and } \mathbb{P}\left(E_2^c\right) \leq \frac{1}{N^4}$$

Define $E \stackrel{\text{def}}{=} E_1 \cap E_2$, then

$$\mathbb{P}\left(E^c\right) \leq \mathbb{P}\left(E_1^c\right) + \mathbb{P}\left(E_2^c\right) \leq \frac{2}{N^4}. \tag{11}$$

Further we define random sets $I_1 \stackrel{\text{def}}{=} \left\{y \in S | n'_y < c_2 \log N, n_y < c_1 \log N \text{ and } \boldsymbol{p}_y \leq \frac{c_1 \log N}{n}\right\}$ and $I_2 \stackrel{\text{def}}{=} \left\{y \in S \mid n'_y > c_2 \log N \text{ and } \boldsymbol{p}_y > \frac{c_3 \log N}{n}\right\}$. We first bound the conditional bias and we later use it to bound the bias of our estimator. Our next statement follows from uniform approximation error [Tim14] and is explicitly written in to Equation 53 of [WY16a].

$$|\mathbb{E}\left[\mathbf{f}_{I_1}(\mathbf{p}) - \hat{g}_{I_1}|I_1\right] = |\sum_{y \in I_1} \boldsymbol{p}_y \log \frac{1}{\boldsymbol{p}_y} - P_{L,g}(\mathbf{p}_y)| \leq \frac{N}{\alpha^2 n \log N}. \tag{12}$$

Let $\hat{\mathbf{p}}$ be the empirical distribution on $x_2^n$. Similarly by analysis of Case 2 and Equation 58 in [WY16a] we have,

$$|\mathbb{E}\left[\mathbf{f}_{I_2}(\mathbf{p}) - \hat{g}_{I_2}|I_2\right] = |\mathbb{E}\left[\sum_{y \in I_2}(\mathbf{p}_y \log\frac{1}{\mathbf{p}_y} - \hat{\mathbf{p}}_y \log\frac{1}{\hat{\mathbf{p}}_y})|I_2\right]| \leq \frac{N}{n \log N} . \quad (13)$$

Combining equations 12, 13, 11 and 10 we can upper bound the bias of our estimator by $\frac{2N}{n \log N} + \frac{4 \log N}{N^4}$. Note here we use the fact that in the case of bad event ($E^c$ or $E_3^c$) the bias of our estimator is upper bounded by $\log N$.

Our analysis for largest change in the value of estimator by changing one sample is exactly the same as [ADOS16] and for completeness we describe it next. The largest coefficient of the optimal uniform polynomial approximation of degree $L$ for function $x \log x$ in the interval $[0, 1]$ is upper bounded by $2^{3L}$. This result follows from the proof of Lemma 2 in [CL11] and is also explicitly mentioned in the proof of Proposition 4 in [WY16a]. Therefore, the largest change (after appropriately normalizing) is the largest value of $b_i$ (co-efficient of the optimal uniform polynomial approximation) which is

$$\frac{2^{3L} e^{L^2/n}}{n}.$$

For $L = 0.25\alpha \log n$, this is at most $\frac{n^\alpha}{n}$. $\qquad \square$

The proof of Lemma 5.1 for entropy follows from the above lemma and Lemma C.1 by substituting $n = O\left(\frac{N}{\log N}\frac{1}{\epsilon}\right)$ and $\epsilon > \Omega\left(\frac{\log N}{N^{1-\alpha}}\right)$.

## C.2  Distance to uniformity

Here we provide proof sketch for the existence of an estimator with desired properties. This result is analogous to Lemma 7 in [ADOS16] and proof for this lemma is very similar to that of [ADOS16] and for completeness we sketch the proof for this result.

**Lemma C.3.** *Let $c_1 > 2c_2$, $c_2 = 35$. Then for all subset $S \subseteq \mathcal{D}$, there is an estimator for distance to uniformity on subset $S$ ($\sum_{y \in S}|\mathbf{p}_y - \frac{1}{N}|$) that changes by at most $n^\alpha/n$ when a sample is changed, and the bias of the estimator is at most $O(\frac{1}{\alpha}\sqrt{\frac{c_1 \log N}{N \cdot n}})$.*

*Proof.* We divide estimation of distance to uniformity into two cases based on $n$. Note the proof for this lemma follows along the lines of [ADOS16].

**Case 1:** $\frac{1}{N} < c_2 \log N/n$. In this case, we use the estimator defined in the last section for $g(x) = |x - 1/k|$.

**Case 2:** $\frac{1}{N} > c_2 \log N/n$. The estimator is as follows for all $y \in S$:

$$g_y = \begin{cases} G_{L,g}(n_y), & \text{for } |\frac{n_y'}{n} - \frac{1}{N}| < \sqrt{\frac{c_2 \log N}{Nn}}, \text{ and } |\frac{n_y}{n} - \frac{1}{N}| < \sqrt{\frac{c_1 \log N}{Nn}}, \\ 0, & \text{for } |\frac{n_y'}{n} - \frac{1}{N}| < \sqrt{\frac{c_2 \log N}{Nn}}, \text{ and } |\frac{n_y}{n} - \frac{1}{N}| \geq \sqrt{\frac{c_1 \log N}{Nn}}, \\ g\left(\frac{n_y}{n}\right), & \text{for } |\frac{n_y'}{n} - \frac{1}{N}| \geq \sqrt{\frac{c_2 \log N}{Nn}}. \end{cases}$$

The estimator proposed in [ADOS16] is exactly the same as ours, but we define our estimator only for the domain elements in $S \subseteq \mathcal{D}$. Note the estimator defined in [JHW17] is slightly different, assigning $G_{L,g}(n_y)$ for the first two cases. As in [ADOS16], this second case is designed to bound the change in value of the estimator by changing one sample. Using [Tim14](Equation 7.2.2), [JHW17] (for their estimator) show that, contribution towards bias (conditioned on "good" event [6]) by any domain element $y \in \mathcal{D}$ (note we only need this result to hold for $y \in S$) satisfying $n_y' < c_2 \log N$, and $n_y <$

$c_1 \log N$ for case 1 and $|\frac{n'_y}{n} - \frac{1}{N}| < \sqrt{\frac{c_2 \log N}{Nn}}$, and $|\frac{n_y}{n} - \frac{1}{N}| < \sqrt{\frac{c_1 \log N}{Nn}}$ for case 2, using polynomial approximation (Lemma 27 in [JHW17]) is upper bounded by $O\left(\frac{1}{L}\sqrt{\frac{\log N}{N \cdot n \log N}}\right)$, where $L$ is the degree of optimal uniform approximation for function $|x - \frac{1}{N}|$ in the interval $[0, 2c_1 \frac{\log N}{N}]$ for case 1 (Equation (351) in [JHW17]) and $[\frac{1}{N} - \sqrt{\frac{c_1 \log N}{Nn}}, \frac{1}{N} - \sqrt{\frac{c_1 \log N}{Nn}}]$ for case 2 (Equation (367) in [JHW17]). Further, the bias (conditioned on the "good" event) by empirical estimate for domain element $y \in \mathcal{D}$ (as before, we need this result to hold only for $y \in S$) satisfying $n'_y \geq c_2 \log N$ for case 1 and $|\frac{n'_y}{n} - \frac{1}{N}| \geq \sqrt{\frac{c_2 \log N}{Nn}}$ for case 2, is zero (Refer proof of Theorem 2 [JHW17]). [JHW17] also bound the probability of "bad" event[6] (Refer proof of Lemma 2 in [JHW17]), thus bounding the bias with respect to these domain elements. Further similar to [ADOS16], by our choice of $c_1, c_2$, the contribution to bias by domain element $y \in S$ satisfying $n'_y < c_2 \log k$, and $n_y \geq c_1 \log N$ for case 1 and $|\frac{n'_y}{n} - \frac{1}{N}| < \sqrt{\frac{c_2 \log N}{Nn}}$, and $|\frac{n_y}{n} - \frac{1}{N}| \geq \sqrt{\frac{c_1 \log N}{Nn}}$ for case 2, is upper bounded by $1/n^4 < \epsilon^2$. Combining analysis of all these cases together, we have our result for bias.

The proof for largest change in the estimator value by changing one sample is exactly same as [ADOS16]. Similar to [ADOS16], here we use the fact [CL11] (Lemma 2) that the largest coefficient of the optimal uniform polynomial approximation of degree $L$ for function $|x|$ in the interval $[-1, 1]$ is upper bounded by $2^{3L}$. $2^{3L}$. Similar to entropy (after appropriate normalization), the largest difference in estimation will be at most $n^\alpha/n$. $\qquad\square$

The proof of Lemma 5.2 for distance to uniformity follows from the above lemma and Lemma C.1 and by substituting $n = O\left(\frac{N}{\log N}\frac{1}{\epsilon^2}\right)$.

## Footnotes

[6]Refer [JHW17] for definitions of "good" and "bad" events.