[Reviews · NeurIPS 2019]

Reviewer 1



The contributions in this paper are quite interesting and possibly significant. The paper fits in an ongoing series of developments for the studied problem, and gives a nice contribution. The main issue with the paper is that it feels extremely dry and it is hard to follow at times. Surely in part this limited readability is due to the relatively high technical level and the page limitation, but more intuition could be given to the reader to help them follow the ongoing reasoning, and to ensure that the paper is at least in part self-contained (for example, some intuition on the proofs of Thm 3.1 and 3.2, and Lemma 4.1 would be helpful).

Reviewer 2



The paper studies two open questions in the PML framework - approximately calculating the PML and showing the optimally of PML based estimation for a larger accuracy range - epsilon. For the problem of support estimation, it is shown that an approximate PML is sample optimal in the entire range of epsilon. For the problem of entropy and distance to uniformity, the authors devise a new generalization of PML, called the Pseudo PML. They do so by defining the S-pseudo profile, using the S-PML estimate for the 'light' elements and the empirical estimate for the 'heavy' elements. Using this, they derive sample optimal rates for these problems in the entire range of epsilon. The results are interesting and resolves some intriguing open questions in [ADOS16]. Pseudo PML is a natural extension of PML and I find the general framework of Algorithm 1 elegant and right in principle. I'm overall satisfied with the writing of the paper, and the authors provide a good sketch of their proofs. I'm also satisfied with the citations of relevant work. I think the paper is interesting and vote to accept the paper. Post Rebuttal: While the authors have addressed some of my concerns, I'm maintaining my score. The authors have to adequately address how PseudoPML is efficiency computable. At least some experimental evaluations that suggest that it is possible to do so.

Reviewer 3



I would like to recommend the paper for acceptance but hesitate due to the following reasons. I think that the paper may need further improvement. 1. Let me begin with a few suggestions/comments on the literature review part (Section 1.1). 1) line 70, "... the empirical distribution is sample competitive", should this be "sample optimal"? 2) line 74, it seems that the sample complexity upper bound obtained [VV11b] is (N/\epsilon^2\log N ) instead of (N/\epsilon\log N ). 3) line 76, maybe it is better to define "u" as "the uniform distribution over D" before using it. 4) line 81, it seems that according to [WY15], one should use [1/N, 1] instead of [1/k, 1]. 5) line 90, [VV11b] does not seem to mention "distance to uniformity" or "support coverage". 6) line 93, the estimator in [HJW18] does not seem to achieve "the optimal sample complexity in all error regimes". According to the last paragraph on page 12 of [HJW18]. The estimator has a sub-optimal standard deviation in all cases and works for \epsilon>n^{-0.33}. 7) line 95, in a recent work of some of the [YOSW18]'s authors, the "n to n\sqrt{\log n}" "data amplification" has been strengthened to "n to n\log n". The new paper is available at "https://arxiv.org/abs/1903.01432", it would be good to also mention this stronger result. 2. The pseudo-PML is novel as the previous PML method always utilizes the full profile, which may not be the best choice for certain problems. The submission also claimed that the method is "efficient" (both in the title and the main body), and "more practical" (abstract, line 9), so I expect to see an efficient algorithm and some experimental results. However, I was not able to find such contents in the current version. 3. I am slightly concerned about the proof of the support-estimation result (Theorem 3.1), which appears in the supplemental material. The proof appears between line 342 and 358. The inequality above line 357 states that given a sample size n>2k/(\log k), the probability that S(p)=Distinct(\phi) is >= 1-exp(n/k). It seems that 1-exp(n/k) is a negative number for any positive integer n, and it is not meaningful to show that a probability is always >= a negative quantity. It is possible that the expression missed a negative sign inside the exponential function. However, even if that is the case, for some n>2k/(\log k), say n=3k/(\log k), the quantity 1-exp(-n/k) is roughly 3/(\log k) for large k, which can be arbitrarily close to 0 as k increases. Hence, we do not have a high-probability guarantee here, e.g., "with probability at least 2/3". 4. The submission seems to suggest that the proposed method is universal and property-independent, e.g., line 101, 188. The detailed construction of these estimators appears in Section 5: Proof of theorem 3.9 and Proof of theorem 3.10. According to the construction, for the properties considered in the paper, namely, entropy and distance to uniformity, one must first choose a specific set F of frequencies according to that property, and then compute the pseudo-PML. Subjectively, I think it may be more accurate to say that this method "weakly depends" on the property. 5. Some terms and notations appeared in the paper before being defined: 1) line 37: "separable" 2) line 72 to 78: "N" 3) line 160 to 161: "F" ===========After rebuttal========== I would like to thank the authors for clarifying the points above. The rebuttal stated that "the key idea of pseudoPML is to use an algorithm ([CSS19]) for PML as a subroutine in a black-box way". Yet I don't see how to use that algorithm to compute a pseudoPML. The paper's title emphasizes that the method is "efficient", I hope that the authors can present solid proofs for this claim. I think that simply "invoke the [CSS19] efficient approximate PML algorithm on frequencies up to O(log N)" will not work here. For example, we can choose two samples, one is of size N and the other is of size N^2, such that the first one only contains distinct symbols, and the second one coincides with the first in its first N entries, and has only one (distinct) symbol in the remaining N^2-N entries. In both cases, invoking "the [CSS19] efficient approximate PML algorithm on frequencies up to O(log N)" will yield the same distribution estimate. Yet it is very likely that the two underlying distributions have quite different entropy values. It is possible that I am missing something, but I do not see it from the paper/rebuttal. For this reason, I would like to keep my original overall score.

[Author Response · NeurIPS 2019]

We thank the reviewers for the helpful comments, great suggestions, and positive feedback. In the final version, we
will be sure to add further intuition about the proofs and insight into parameter choices to clarify the presentation.
Further, we ran multiple experiments (detailed below) corroborating our theoretical claims. Finally, we thank reviewer
3 for pointing out that the result of [HJW18] holds only when $\epsilon > n^{-0.33}$ and therefore our result outperforms all
previous universal estimators when applied to estimating entropy and distance to uniformity for $\epsilon < n^{-0.33}$. We hope
the collection of proposed changes and new experiments elevate your view of the paper.

**Experiments:** We performed two sets of experiments. First, we compared the error of multiple estimators for computing
entropy (in the full version we will repeat these experiments for distance to uniformity and more distributions).

Each plot depicts the performance of various algorithms for estimating entropy of different distributions with domain
size $N = 10^5$. "Mix 2 Uniforms" is a mixture of two uniform distributions, with half the probability mass on the first
$N/10$ symbols, and $\text{Zipf}(\alpha) \sim 1/i^\alpha$ with $i \in [N]$. MLE is the naive approach of using the empirical distribution with
correction bias, [PJW17] is arXiv:1712.07177; all the remaining are cited in our paper. Each data point represents 50
random trials. In our algorithm we pick $threshold = 18$ (same as [WY16]) and our set F $= [0, 18]$ (input of algorithm
1), meaning, we run PML on frequencies $\leq 18$ and empirical on the rest. We use the heuristic algorithm in [PJW17] to
compute an approximate PML distribution. Our results are competitive with other state of the art entropy estimators.

Second, to demonstrate the practicality of our approach, we compare the running time of our algorithm using [PJW17]
as a subroutine to the raw [PJW17] algorithm on the $\text{Zipf}(1)$ distribution. The second row is the fraction of samples on
which our algorithm performs empirical estimate (plus correction bias). The third row is the ratio of running time of
[PJW17] to our algorithm. For larger sizes we get $\geq 10$x speedup.

| Samples size | $10^3$ | $5*10^3$ | $10^4$ | $5*10^4$ | $10^5$ | $5*10^5$ | $10^6$ | $5*10^6$ | $10^7$ |
|---|---|---|---|---|---|---|---|---|---|
| EmpFrac | 0.18382 | 0.31654 | 0.37150 | 0.50457 | 0.56239 | 0.69533 | 0.75245 | 0.88554 | 0.94282 |
| Speedup | 0.824 | 1.205 | 1.669 | 3.561 | 4.852 | 9.552 | 13.337 | 12.196 | 10.204 |

**Reviewer 1:** Thank you! As discussed above we will be sure to add more intuition in the final version.

**Reviewer 2.** Thank you! Comparing the relative strengths of pseudoPML versus PML is an interesting direction for
future research. One advantage of pseudoPML is that it is a simple approach that facilitates provable guarantees for a
broader range of parameter $\epsilon$. For simulations see above and further we will address all the writing related suggestions.

**Reviewer 3:** Thank you! See below for response to detailed comments and beginning for response to improvements.

**Point 1, 5.** Great points; We agree and will address all in the final verison.

**Point 2.** Beyond the experiments discussed above, we note that the key idea of pseudoPML is to use an algorithm for
PML as a subroutine in a black box way. For example, for entropy if we invoke the [CSS19] efficient approximate PML
algorithm on frequencies up to $O(\log N)$, then our pseudo-PML algorithm gives a nearly linear time sample-optimal
estimator whenever $\epsilon > N^{-0.33}$, improving upon [CSS19] which applied only when $\epsilon > N^{-0.199}$. Similarly for
distance to uniformity if we invoke the [CSS19] efficient approximate PML algorithm on $O(\sqrt{n \log n/N})$ distinct
frequencies, then our pseudo-PML algorithm gives a nearly linear time sample-optimal estimator whenever $\epsilon > N^{-0.33}$,
improving upon [CSS19] which applied only when $\epsilon > N^{-0.249}$.

**Point 3.** In the proof of Theorem 3.1, the inequality above line 357 should be $\Pr(S(p) = \text{Distinct}(\phi)) >= 1 -$
$k \exp(-n/k)$. Since $\epsilon > 1/k^{0.249}$ is handled in [ADOS16], we consider $\epsilon \leq 1/k^{0.249}$. Using sample size $n =$
$\Theta(k \log^2(1/\epsilon)/\log k)$, we have $n \geq ck \log k$ for constant $c \geq 2$. Combining this with $\Pr(S(p_\phi) = \text{Distinct}(\phi)) = 1$
(where $p_\phi$ is the PML) we have that our success probability, $\Pr(S(p_\phi) = S(p)) \geq 1 - k \exp(-n/k) \geq 1 - \exp(-\log k)$.

**Point 4.** We agree that it is more accurate to say our method "weakly depends" on the property and we will update.

[Meta-Review · NeurIPS 2019]

The reviews were mixed but all the reviewers found the paper interesting and relevant. Some of their concerns about the paper were handled by the authors in the rebuttal. The authors are encouraged to improve the camera-ready paper following the recommendations of the reviewers, especially to give a better motivation and explanation for PseudoPML. Furthermore, readability of the paper will be much improved via examples. This meta-review was reviewed and revised by the Program Chairs, based on discussions with the Senior Area Chair.